# Magnesium Supplementation Attenuates Ultraviolet-B-Induced Damage Mediated through Elevation of Polyamine Production in Human HaCaT Keratinocytes

**DOI:** 10.3390/cells11152268

**Published:** 2022-07-22

**Authors:** Shokoku Shu, Mao Kobayashi, Kana Marunaka, Yuta Yoshino, Makiko Goto, Yuji Katsuta, Akira Ikari

**Affiliations:** 1Laboratory of Biochemistry, Department of Biopharmaceutical Sciences, Gifu Pharmaceutical University, Gifu 501-1196, Japan; 165041@gifu-pu.ac.jp (S.S.); 155032@gifu-pu.ac.jp (M.K.); 136033@gifu-pu.ac.jp (K.M.); yoshino-yu@gifu-pu.ac.jp (Y.Y.); 2Shiseido Co., Ltd., MIRAI Technology Institute, Yokohama 220-0011, Japan; makiko.goto@shiseido.com (M.G.); yuji.katsuta@shiseido.com (Y.K.)

**Keywords:** polyamine synthase, UVB, magnesium

## Abstract

Magnesium ions (Mg^2+^) have favorable effects such as the improvement of barrier function and the reduction of inflammation reaction in inflammatory skin diseases. However, its mechanisms have not been fully understood. Microarray analysis has shown that the gene expressions of polyamine synthases are upregulated by MgCl_2_ supplementation in human HaCaT keratinocytes. Here, we investigated the mechanism and function of polyamine production. The mRNA and protein levels of polyamine synthases were dose-dependently increased by MgCl_2_ supplementation, which were inhibited by U0126, a MEK inhibitor; CHIR-99021, a glycogen synthase kinase-3 (GSK3) inhibitor; and Naphthol AS-E, a cyclic AMP-response-element-binding protein (CREB) inhibitor. Similarly, reporter activities of polyamine synthases were suppressed by these inhibitors, suggesting that MEK, GSK3, and CREB are involved in the transcriptional regulation of polyamine synthases. Cell viability was reduced by ultraviolet B (UVB) exposure, which was rescued by MgCl_2_ supplementation. The UVB-induced elevation of reactive oxygen species was attenuated by MgCl_2_ supplementation, which was inhibited by cysteamine, a polyamine synthase inhibitor. Our data indicate that the expression levels of polyamine synthases are upregulated by MgCl_2_ supplementation mediated through the activation of the MEK/GSK3/CREB pathway. MgCl_2_ supplementation may be useful in reducing the UVB-induced oxidative stress in the skin.

## 1. Introduction

The important roles of skin are to maintain body hydration, temperature, moisture, and sensations [1]. Exposure of skin to high doses of ultraviolet (UV) can induce initiating events of skin inflammation and carcinogenesis. The UV spectrum is divided into three major sections: short-wave UV with 250–290 nm (UVC), middle-wave UV with 290–320 nm (UVB), and long-wave UV with 320–400 nm (UVA). UVC is blocked by the ozone layer, but UVA and UVB can reach the Earth’s surface and fall on the skin, eyes, and hair of humans. UVA irradiation is preferentially absorbed by lipid rafts of cell membrane and induces the formation of singlet oxygen. UVB irradiation is absorbed by DNA and proteins, resulting in the production of reactive molecule species. High doses of UVB (>25 mJ/cm^2^) show a cytotoxic effect in various cells including keratinocytes. The UVB-induced cell damage is caused by the generation of reactive oxygen species (ROS) including superoxide anion radical, hydroxy radical, and hydrogen peroxide [2].

Magnesium is a divalent cation most abundantly existing in the human body, and a majority of it is stored in the bone, muscle, and liver. Magnesium ions (Mg^2+^) play a pivotal role in the regulation of hundreds of enzymatic reactions including synthesis of DNA, RNA, and protein; glucose metabolization; and energy production [3]. In the skin, Mg^2+^ has functions such as acceleration of tissue repair [4] and suppression of inflammatory response [5]. Bathing in a Mg^2+^-rich Dead Sea salt solution induces several favorable effects, including the improvement of skin barrier function, enhancement of skin hydration, and reduction of inflammation in atopic dry skin [6]. These effects may be caused by the properties of Mg^2+^ such as water binding capability. However, the functional mechanism of Mg^2+^ has not been fully understood.

Three major polyamines (putrescine, spermidine, and spermine) are present in living cells, and spermine is especially abundantly contained in the human epidermis [7]. These polyamines play pivotal roles in cell proliferation, differentiation, apoptosis, and protection from oxidative damage [8,9]. Putrescine, which is synthesized from ornithine by ornithine decarboxylase (ODC), is converted to spermidine and spermine by the addition of aminopropyl groups derived from decarboxylated S-adenosylmethionine, which is converted from S-adenosylmethionine (SAM) to S-adenosyl-methioninamine (dcSAM) by adenosylmethionine decarboxylase 1 (AMD1) (Figure 1). These reactions are catalyzed by spermidine synthase (SRM) and spermine synthase. Intracellular polyamine contents are strictly regulated by a combination of synthesis, degradation, uptake, and excretion. The mice targeted disruption of *ODC* and *AMD1* genes died early in embryonic development [10,11]. The level of spermidine in skin changes during aging peaked at 10 weeks old and was markedly reduced at 26 weeks old [12]. Inflammation and macrophage activation in mouse skin are inhibited by natural polyamines [13]. In addition, polyamines can function as biological antioxidants related to radical scavenging activity [14].

In the present study, DNA microarray analysis showed that the expression levels of polyamine synthases are upregulated by MgCl_2_ supplementation in human HaCaT keratinocytes. Therefore, we investigated the mechanism and function of MgCl_2_ supplementation-induced polyamine production. The mRNA and protein levels were examined using real-time polymerase chain reaction (PCR) and Western blotting analyses, respectively. The reporter activities of polyamine synthases were assessed using luciferase assay. The production of ROS and lipid peroxide was measured using specific fluorescent probes. Our results indicate that MgCl_2_ supplementation may be involved in the protection on HaCaT cells against UVB-induced damage mediated through the production of polyamines.

## 2. Materials and Methods

### 2.1. Cell Culture

HaCaT cells, an immortalized human keratinocyte line [15], were grown in Dulbecco’s modified Eagle’s medium (DMEM, 044-29765, Fujifilm Wako Pure Chemical, Osaka, Japan) supplemented with 5% fetal bovine serum (FBS, F7524, Sigma-Aldrich, St. Louis, MO, USA), 0.07 mg/mL penicillin-G potassium, and 0.14 mg/mL streptomycin sulfate in a 5% CO_2_ atmosphere at 37 °C. Mg^2+^-free medium was prepared according to the composition of normal DMEM (0.8 mM Mg^2+^) without Mg^2+^. Normal skin human keratinocyte NHEK/SVTERT3-5 cells (CLHT-011-0026) were cultured according to the instructions provided by Evercyte GmbH (Vienna, Austria). The cells were treated with various concentrations of Mg^2+^ in the FBS-free DMEM to avoid the effect of growth factors in serum. Cell viability was measured using a Cell Counting Kit-8 (CCK-8, CK04, Dojindo Laboratories, Kumamoto, Japan).

### 2.2. DNA Microarray Analysis

Cells were inoculated at a density of 1 × 10^5^/dish on 6 cm dishes and cultured for 96 h. Total RNA was extracted using NucleoSpin RNA (740955.10, Takara Bio, Shiga, Japan). DNA microarray analysis was performed using GeneChip Human Gene 2.0 ST array (a contract analysis service at Filgen Incorporation Japan, Nagoya, Japan). The cutoff point was set to twofold.

### 2.3. Extraction of Total RNA and Quantitative Real-Time PCR

Cells were inoculated at a density of 7 × 10^4^/well on 6-well plates and cultured for 96 h. Total RNA was extracted using TRI reagent (TR118, Molecular Research Center, Cincinnati, OH, USA). Reverse transcription and quantitative real-time PCR reactions were performed using a ReverTraAce qPCR RT Kit (FSQ-101, Toyobo, Osaka, Japan) and a Thunderbird SYBR qPCR Mix (QPS-201, Toyobo), respectively, as described previously [16]. Primer pairs used in real-time PCR are shown in Table 1.

### 2.4. SDS-Polyacrylamide Gel Electrophoresis and Western Blotting

Cells were inoculated at a density of 7 × 10^4^/well on 6-well plates and cultured for 96 h. The cells were collected in cold phosphate-buffered saline and precipitated by centrifugation. Cell lysates were prepared as described previously [17]. The aliquots (30–50 μg) were applied to SDS-polyacrylamide gel electrophoresis and Western blotting. Rabbit anti-AMD1 (11052-1-AP), anti-GSK3β (22104-1-AP), and anti-SRM (15979-1-AP) antibodies were obtained from ProteinTech (Tokyo, Japan). Goat anti-β-actin (sc-1615) and rabbit anti-p-ERK1/2 (sc-16982-R) antibodies were from Santa Cruz Biotechnology (Santa Cruz, CA, USA). Rabbit anti-ERK1/2 (#4695) and anti-p-GSK3β (#5558) antibodies were from Cell Signaling Technologies (Beverly, MA, USA). These primary antibodies were used at the dilution 1:1000. The blotted membrane was scanned with a C-DiGit Blot Scanner (LI-COR Biotechnology, Lincoln, NE, USA).

### 2.5. Measurement of Polyamine Contents

Cells were inoculated at a density of 5 × 10^3^/well on 96-well plates and cultured for 96 h. The production of polyamines was measured using an intracellular polyamine detection reagent, PolyamineRED (FDV-0020, Biofunctional Synthetic Chemistry Laboratory, Cluster for PioneeringResearch, RIKEN, Saitama, Japan). The nucleus was stained with 4′,6-diamidino-2-phenylindole (DAPI, D212, Dojindo Laboratories). The fluorescence images were obtained using a BZ-X800 fluorescence microscope (Keyence, Osaka, Japan), and the fluorescence intensities of PolyamineRED were calculated by ImageJ software.

### 2.6. Transfection of Reporter Vector

Cells were inoculated at a density of 2 × 10^4^/well on 24-well plates. After 24 h, the GLuc-ON Promoter Reporter vectors (0.8 μg/well) for human *SRM* (HPRM44738-PG02, GeneCopoeia, Rockville, MD, USA) and *AMD1* (HPRM41795-PG02, GeneCopoeia) were transfected into the cells using HilyMax (H357, Dojindo Laboratories). Transfection efficiency was compensated by secreted alkaline phosphatase (SEAP) reporter gene (0.2 μg/well). The activities of luciferase and SEAP were measured using a Ready-To-Glow Dual Secreted Reporter Assay kit (631734, Takara Bio).

### 2.7. Immunocytochemistry

Cells were inoculated at a density of 7 × 10^4^/well on 6-well plates containing sterile cover glasses and were cultured for 96 h. After fixation, permeabilization, and blocking, the cells were incubated with mouse anti-p-CREB (S133) antibody (1:100 dilution, R&D Systems, Minneapolis, MN, USA) followed by incubation with donkey anti-mouse Alexa 488 monoclonal antibody (1:100, Thermo Fisher Scientific, Waltham, MA, USA) and DAPI. The fluorescence images were obtained using an LSM700 confocal laser microscope (Carl Zeiss, Jena, Germany), as described previously [16], and the fluorescence intensities were calculated by ImageJ software.

### 2.8. Generations of ROS and Lipid Peroxides

Cells were inoculated at a density of 5 × 10^3^/well on 96-well plates and cultured for 96 h. The generation of ROS and lipid peroxides was detected by a 2′,7′-dichlorodihydrofluorescein-diacetate (DCF-DA, D399, Thermo Fisher Scientific) and Liperfluo, a fluorescence dye used for specific detection of lipid peroxides (L248, Dojindo Laboratories), respectively. The fluorescence intensity was measured using an Infinite F200 fluorescence microplate reader (Tecan, Mannedorf, Switzerland).

### 2.9. Statistics

Results are presented as means ± SEM. Differences between groups were analyzed using one-way analysis of variance, and corrections for multiple comparison were made using Tukey’s multiple comparison test. Comparisons between two groups were made using Student’s *t*-test. Statistical analyses were performed using KaleidaGraph version 4.5.1 software (Synergy Software, Reading, PA, USA). Significant differences were assumed at *p* < 0.05.

## 3. Results

### 3.1. Effect of Extracellular MgCl_2_ Concentration on Gene Expression in HaCaT Cells

HaCaT cells were exposed to the media containing 0.8 mM MgCl_2_ (normal concentration)- or 5.8 mM MgCl_2_ (high concentration)-containing media for 6 h. DNA microarray analysis using total RNA isolated from HaCaT cells showed that 45 genes are upregulated and 13 genes are downregulated by high MgCl_2_ concentration (Table 2, Appendix A). All 58 differentially expressed genes (DEGs) were analyzed by DAVID software, and the results of GO analysis indicated that (1) for metabolic process (BP), upregulated DEGs were particularly enriched in DNA replication, DNA replication initiation, DNA replication checkpoint, and mitotic DNA replication checkpoint, and downregulated DEGs in cellular calcium ion homeostasis; (2) for molecular function (MF), upregulated DEGs were enriched in DNA replication origin binding, DNA binding, and enzyme binding; (3) for cell component (CC), upregulated DEGs were enriched in nucleoplasm and nucleus, and downregulated DEGs in nuclear membrane (Appendix A). KEGG analysis demonstrated that upregulated DEGs were particularly enriched in cell cycle, DNA replication, and arginine and proline metabolism, whereas downregulated DEGs were not significantly enriched in any signaling pathways (Appendix A).

We selected two types of spermidine synthesis-related genes, *SRM* and *AMD1*, which were upregulated in 5.8 mM MgCl_2_-containing media. To validate the data of microarray, we performed real-time PCR with specific primers to *SRM* and *AMD1*. The mRNA levels of *SRM* and *AMD1* were dose-dependently increased by MgCl_2_ supplementation in HaCaT cells, and the effects were significant over the concentration of 3.3 mM MgCl_2_ (Figure 2A). Similarly, the mRNA levels of *SRM* and *AMD1* were significantly increased by Mg-lactate and MgSO_4_ supplementation (Figure 2B). In addition, the mRNA levels of *SRM* and *AMD1* were increased by MgCl_2_ supplementation in NHEK/SVTERT3-5 cells (Figure 2C). These results indicate that both *SRM* and *AMD1* expressions may be upregulated by high Mg^2+^ concentration in human keratinocytes. In contrast, neither *SRM* nor *AMD1* expressions are significantly changed by MgCl_2_ depletion in HaCaT cells (Figure 2D).

### 3.2. Effect of MgCl_2_ Supplementation on Polyamine Synthesis

HaCaT cells were exposed to the media containing 0.8, 3.3, 5.8, and 10.8 mM MgCl_2_ for 24 h. The protein levels of SRM and AMD1 were dose-dependently increased by MgCl_2_ supplementation (Figure 3A). These results are similar to those in real-time PCR analysis. Spermidine is a polyamine compound that is present in the cytoplasm of eukaryotic cells [18]. The production of intracellular polyamines can be detected using a fluorescent probe, PolyamineRED. The fluorescence intensity of PolyamineRED was dose-dependently increased by MgCl_2_ supplementation, and the relative values were 157.3 ± 8.6% (5.8 mM MgCl_2_) and 183.8 ± 16.2% (10.8 mM MgCl_2_) (Figure 3B). These results indicate that MgCl_2_ supplementation may upregulate the production of polyamines mediated by the elevation of spermidine-synthesis-related enzymes.

### 3.3. Effects of Intracellular Signaling Pathway Inhibitors on SRM and AMD1 Expressions

We recently reported that MgCl_2_ supplementation can activate a GSK3β/CREB pathway in HaCaT cells [17]. To clarify the regulatory mechanisms of SRM and AMD1 expressions, we investigated the effects of inhibitors for intracellular signaling cascades. MgCl_2_ supplementation upregulated the mRNA levels of *SRM* and *AMD1*, which were completely blocked by CHIR99021, a GSK3 inhibitor, and Naphthol AS-E, a CREB inhibitor, in HaCaT cells (Figure 4A). The activity of GSK3 could be regulated by p38/MSK1 and MEK/ERK pathways [19,20]. The MgCl_2_ supplementation-induced elevation of *SRM* and *AMD1* expressions were inhibited by U0126, a MEK inhibitor, but not by apigenin, a MSK1 inhibitor (Figure 4B). These results indicate that MgCl_2_ supplementation may activate GSK3 mediated by the MEK/ERK pathway in HaCaT cells.

### 3.4. Effect of MgCl_2_ Supplementation on Intracellular Signaling Pathways

The effects of inhibitors against intracellular signaling pathways were investigated by Western blotting. The p-ERK1/2 level was increased by MgCl_2_ supplementation, which was inhibited by U0126, but not by CHIR99021, in HaCaT cells (Figure 5A). On the other hand, the p-GSK3β level was increased by MgCl_2_ supplementation, which was inhibited by both U0126 and CHIR99021. Active form p-CREB is translocated from the cytosol to the nucleus and activates the transcription of target genes [21]. The fluorescence intensity of p-CREB in the nuclei was increased by MgCl_2_ supplementation, which was inhibited by U0126, CHIR99021, and Naphthol AS-E (Figure 5B). These results indicate that the nuclear localization of p-CREB may be upregulated by a MEK/ERK/GSK3β pathway in HaCaT cells.

### 3.5. Effect of MgCl_2_ Supplementation on Reporter Activities of SRM and AMD1

HaCaT cells were transiently transfected with *SRM* or *AMD1* reporter plasmid plus the internal pSEAP2 control plasmid. The reporter activities of *SRM* and *AMD1* were significantly increased by MgCl_2_ supplementation compared with those in control medium (0.8 mM MgCl_2_), and the values were 142.5 ± 13.2% (*SRM*) and 136.6 ± 2.5% (*AMD1*). The effects of MgCl_2_ supplementation were significantly inhibited by U0126, CHIR99021, and Naphthol AS-E (Figure 6). These results are similar to those in real-time PCR and Western blotting analyses, indicating that the expressions of SRM and AMD1 may be upregulated by MgCl_2_ supplementation at the transcriptional level.

### 3.6. Protection for UVB- and H_2_O_2_-Induced Cell Damages by MgCl_2_ Supplementation

The viabilities of HaCaT cells were reduced by the treatments with UVB and H_2_O_2_ for 24 h, and the relative values were 60.4 ± 1.9% (100 mJ/cm^2^ UVB), 35.3 ± 0.7% (150 mJ/cm^2^ UVB), 54.9 ± 1.3% (250 mM H_2_O_2_), and 20.2 ± 2.3% (500 μM H_2_O_2_) (Figure 7A). The UVB- and H_2_O_2_-induced cell damages were significantly rescued by preincubation with 5.8 or 10.8 mM MgCl_2_. Similar effects were observed by preincubation with 100 μM or 200 μM spermidine (Figure 7B). To clarify the involvement of spermidine, the effect of cysteamine, a spermidine synthase inhibitor, was examined. The UVB- and H_2_O_2_-induced cell damages were significantly rescued by preincubation with 10.8 mM MgCl_2_, which were inhibited by the cotreatment with cysteamine (Figure 7C). These results indicate that MgCl_2_ supplementation may alleviate the UVB- and H_2_O_2_-induced cell damages mediated through the production of spermidine in HaCaT cells.

### 3.7. Inhibition of UVB-Induced Production of ROS and Lipid Peroxide by MgCl_2_ Supplementation

The production of ROS and lipid peroxide in HaCaT cells was detected using fluorescent probes DCF and Liperfluo, respectively. The fluorescence intensity of DCF was significantly increased by UVB treatment, and the relative value was 124.6 ± 4.8% (Figure 8A). The UVB-induced elevation of ROS production was significantly inhibited by preincubation with 10.8 mM MgCl_2_, spermidine, or *N*-acetyl-cysteine (NAC). The effect of MgCl_2_ supplementation was significantly blocked by the cotreatment with cysteamine. Similar results were observed in the measurement of Liperfluo (Figure 8B). These results indicate that MgCl_2_ supplementation may attenuate ROS-induced cell damage mediated through the production of spermidine in HaCaT cells.

## 4. Discussion

Magnesium is essential for the maintenance of physiological homeostasis in the biological systems of animals. DNA microarray and KEGG pathway analyses in HaCaT cells showed that high MgCl_2_ supplementation upregulates the expression levels of genes involved in cell cycle, DNA replication, and arginine and proline metabolism (Appendix A). Mg^2+^ plays a pivotal role in the regulation of enzymatic reactions including synthesis of DNA, RNA, and protein [3]. Therefore, it is natural that the genes involved in cell cycle and DNA replication were upregulated. Notably, we found that high MgCl_2_ supplementation is involved in the regulation of arginine and proline metabolism in HaCaT cells.

Arginine is used for production of polyamines, nitric oxide, creatine, and urea. Polyamines play important roles in the maintenance of physiological function of skin. The production of polyamines is regulated by several enzymes including ODC, AMD1, and SRM (Figure 1). MgCl_2_ supplementation upregulates the expression levels of SRM and AMD1 (Figure 2A), as well as the production of polyamines (Figure 3B). The promoter activity of ODC is upregulated by ERK1/2 and nuclear factor κB [22], CREB [23], and Sp1 [24], whereas it is downregulated by p38 MAPK [25] and Sp3 [24]. The literature concerning regulatory mechanisms of AMD1 and SRM are very limited. The expression level of AMD1 is upregulated by androgen [26], insulin [27], and polyamines [28]. The expression level of SRM is upregulated by dcSAM, a decarboxylated product of SAM catalyzed by AMD1. Here, we found that MgCl_2_ supplementation upregulates the expression levels of SRM and AMD1 mediated by the activation of MEK/ERK/GSK3β/CREB pathway in HaCaT cells (Figure 4, Figure 5 and Figure 6). The elevation of *SRM* and *AMD1* mRNA levels were also observed in the supplementation of Mg-lactate and MgSO_4_ (Figure 2B). Therefore, the elevation of Mg^2+^ concentration is suggested to be involved in the upregulation of these genes. At present, it is unknown as to how high Mg^2+^ concentration can activate the signaling pathway. Mg^2+^ has dual roles in the regulation of protein kinase reactions [29]. One is the nucleotide substrate such as MgATP. Another is the enzyme/metal-nucleotide complex, leading to the elevation of catalytic efficiency of enzymes. High Mg^2+^ concentration affects steady-state kinetic parameters of various receptor and non-receptor protein kinases. The catalytic efficiency of ERK2 is increased by high Mg^2+^ concentration [30]. The change of catalytic efficiency may be involved in the reaction caused by MgCl_2_ supplementation in HaCaT cells.

The promoter region of the *AMD1* gene contains the binding sites for transcriptional factors including AP-1, AP-2, CREB, SP-1, and multiple steroid receptors [31]. In contrast, there is no report concerning transcriptional factors of the *SRM* gene. The reporter vector of SRM used in the present study contains 1508 bp of promoter region of human *SRM* (HPRM44738). Binding site analysis with the TFBIND program reveals one putative CREB-binding site at the residues-1040/-1029 in the promoter region of *SRM*. The reporter activities of AMD1 and SRM were increased by MgCl_2_ supplementation, which were completely inhibited by Naphthol AS-E, a CREB inhibitor (Figure 6). CREB may function as transcriptional factor in MgCl_2_ supplementation-induced elevation of AMD1 and SRM.

UVB irradiation to the skin of hairless mice induces the elevation of ODC activity, as well as generations of putrescine and spermidine [32]. However, the function of these polyamines has not been fully understood. The exposure of HaCaT cells to UVB (>100 mJ/cm^2^) and H_2_O_2_ (>250 μM) induced cell damage, which was partially rescued by pre-incubation of high concentration of MgCl_2_ (Figure 7A). In addition, the effect of MgCl_2_ supplementation was canceled by cysteamine, a polyamine, and spermidine is able to protect the cells against UVB- and H_2_O_2_-induced damage (Figure 7B,C). These results suggest that the elevation of polyamine levels is involved in the MgCl_2_ supplementation-induced protection in HaCaT cells. Similarly, the cytoprotective effects of polyamines are reported in human skin fibroblasts [33]. Polyamines have various biological functions, including anti-inflammatory and antioxidant properties [8,9]. The antioxidant activity of spermine is over 30-fold that of vitamin E [34]. The UVB-induced generations of ROS and lipid peroxide were suppressed by the pretreatment of high Mg^2+^- or spermidine-containing media (Figure 8). Lipid peroxide induces a free radical chain reaction mechanism followed by elevation of lipid peroxidation. The production of lipid peroxidation in the human skin is increased by chronic sun light exposure and advancing age [35]. Lipid peroxidation is harmful to epidermal cells because it leads to a decrease in the cell viability [36] and induction of inflammatory response [37]. Therefore, the MgCl_2_ supplementation may exert protective effects against UVB mediated through the production of polyamines and their antioxidative properties. Many antioxidants, including superoxide dismutase [38], glutathione reductase [39], and vitamin E [40], are destroyed by UV irradiation in the skin. MgCl_2_ supplementation may be useful to maintain antioxidant capacity in the skin.

In conclusion, we found that the expressions of the spermidine-synthesis-related enzymes SRM and AMD1 are increased by a high concentration of Mg^2+^ in HaCaT cells. The transcriptional activities of these genes were upregulated by the MSK1/GSK3/CREB pathway. The UVB- and H_2_O_2_-induced cell damages were attenuated by the pretreatment of high Mg^2+^- or spermidine-containing media. These results suggest that Mg^2+^ supplementation is useful to enhance barrier function against UVB and ROS damages.

## Figures and Tables

**Figure 1 cells-11-02268-f001:**
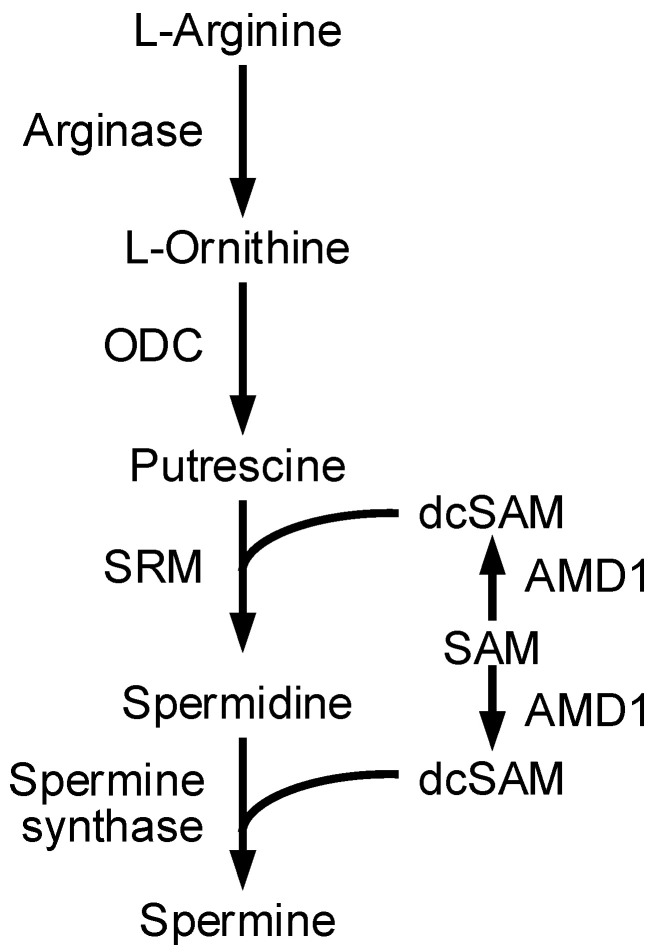
Polyamine biosynthesis pathway.

**Figure 2 cells-11-02268-f002:**
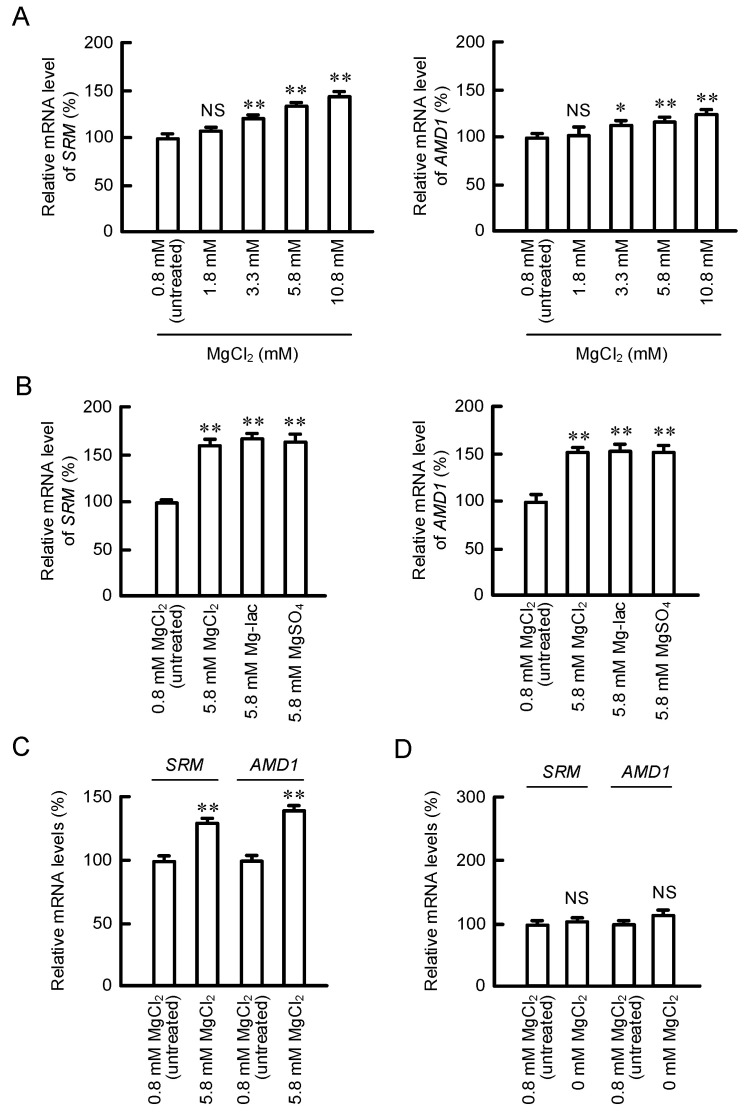
Increases in *SRM* and *AMD1* mRNA levels by MgCl_2_ supplementation. (**A**) HaCaT cells were incubated in the presence of 0.8, 1.8, 3.3, 5.8, and 10.8 mM MgCl_2_ for 6 h. The mRNA levels of *SRM* and *AMD1* were measured by real-time PCR analysis and represented as a percentage of 0.8 mM MgCl_2_. (**B**,**D**) HaCaT cells were incubated in the presence of 0, 0.8, and 5.8 mM MgCl_2_; 5.8 mM Mg-lactate (Mg-lac); or 5.8 mM MgSO_4_ for 6 h. (**C**) NHEK/SVTERT3-5 cells were incubated in the presence of 0.8 and 5.8 mM MgCl_2_ for 6 h. The mRNA levels of *SRM* and *AMD1* were represented as a percentage of 0.8 mM MgCl_2_. n = 3–4. Error bars indicate SEM. ** *p* < 0.01 and * *p* < 0.05 significantly different from 0.8 mM MgCl_2_. ^NS^ *p* > 0.05.

**Figure 3 cells-11-02268-f003:**
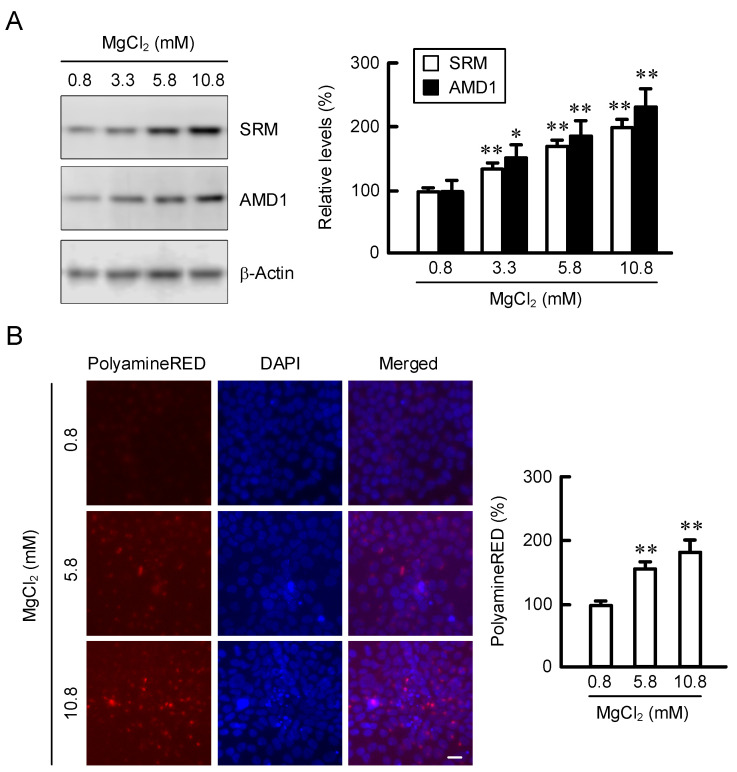
Increase in polyamine production by MgCl_2_ supplementation. (**A**) Cells were incubated in the presence of 0.8, 3.3, 5.8, and 10.8 mM MgCl_2_ for 24 h. The protein levels of SRM and AMD1 were measured by Western blotting analysis and represented as a percentage of 0.8 mM MgCl_2_. (**B**) Cells were incubated in the presence of 0.8, 5.8, and 10.8 mM MgCl_2_ for 24 h, followed by incubation with PolyamineRED and DAPI for 15 min. The fluorescence images were taken using a fluorescence microscope. The fluorescence intensity of PolyamineRED was calculated by ImageJ software and represented as a percentage of 0.8 mM MgCl_2_. Scale bar indicates 10 μm. n = 3–6. Error bars indicate SEM. ** *p* < 0.01 and * *p* < 0.05 significantly different from 0.8 mM MgCl_2_.

**Figure 4 cells-11-02268-f004:**
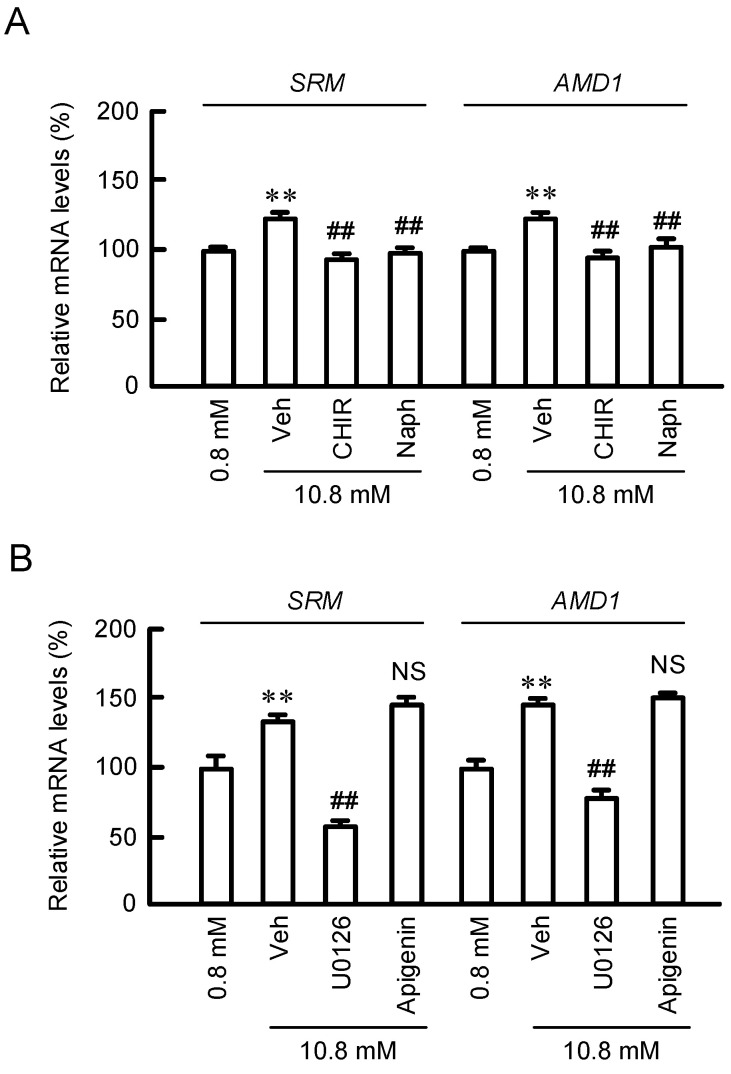
Effects of inhibitors for intracellular signaling factors on *SRM* and *AMD1* mRNA levels. (**A**,**B**) Cells were pre-incubated in the absence (Veh) and presence of 10 μM CHIR99021 (CHIR) or 10 μM Naphthol AS-E (Naph) for 0.5 h. Then, the cells were incubated with 0.8 or 10.8 mM MgCl_2_ for 6 h. The mRNA levels of *SRM* and *AMD1* were represented as a percentage of 0.8 mM MgCl_2_. (**B**) Cells were pre-incubated in the absence (Veh) and presence of 10 μM U0126 or 10 μM apigenin for 0.5 h. Then, the cells were incubated with 0.8 or 10.8 mM MgCl_2_ for 6 h. The mRNA levels of *SRM* and *AMD1* were represented as a percentage of 0.8 mM MgCl_2_. n = 3–4. Error bars indicate SEM. ** *p* < 0.01 significantly different from 0.8 mM MgCl_2_. ^##^ *p* < 0.01 significantly different from Veh. ^NS^ *p* > 0.05.

**Figure 5 cells-11-02268-f005:**
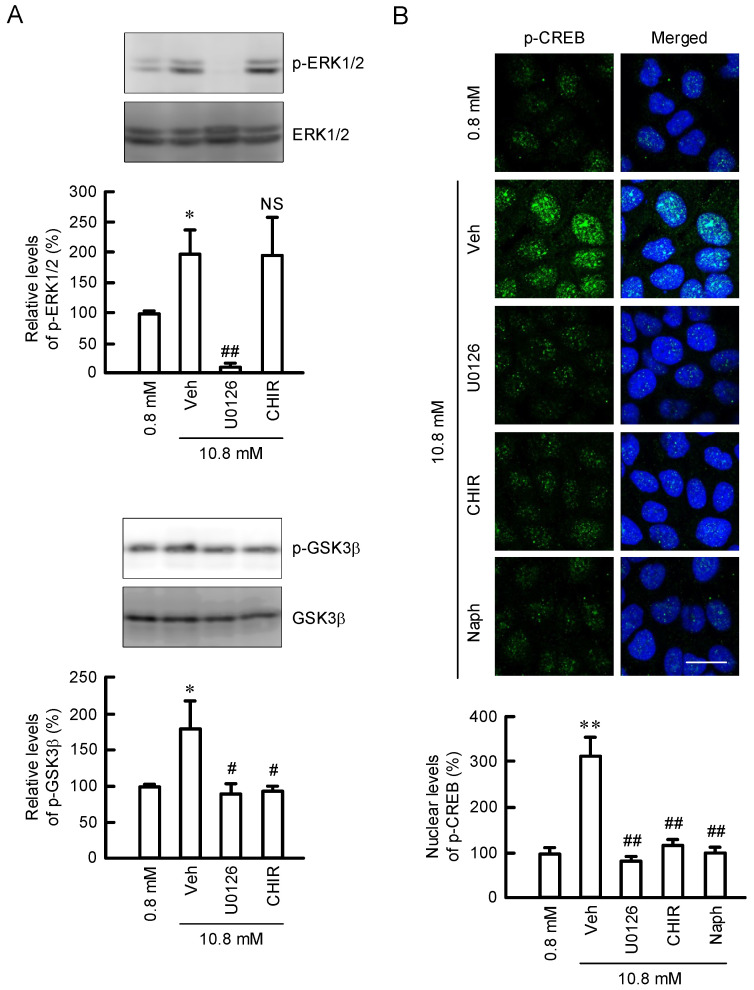
Phosphorylation of ERK1/2, GSK3β, and CREB by MgCl_2_ supplementation. (**A**) Cells were pre-incubated in the absence (Veh) and presence of 10 μM U0126 or 10 μM CHIR99021 (CHIR) for 0.5 h. Then, the cells were incubated with 0.8 or 10.8 mM MgCl_2_ for 1 h. The protein levels of p-ERK1/2 and p-GSK3β were corrected by ERK1/2 and GSK3β, respectively, and represented as a percentage of 0.8 mM MgCl_2_. (**B**) Cells were pre-incubated in the absence (Veh) and presence of 10 μM U0126, 10 μM CHIR, or 10 μM Naphthol AS-E (Naph) for 0.5 h. Then, the cells were incubated with 0.8 or 10.8 mM MgCl_2_ for 1 h. The fluorescence images were taken using a fluorescence microscope. The fluorescence intensity was calculated by ImageJ software and represented as a percentage of 0.8 mM MgCl_2_. Scale bar indicates 20 μm. n = 3–4. Error bars indicate SEM. ** *p* < 0.01 and * *p* < 0.05 significantly different from 0.8 mM MgCl_2_. ^NS^ *p* > 0.05. ^##^ *p* < 0.01, and ^#^ *p* < 0.05 significantly different from Veh.

**Figure 6 cells-11-02268-f006:**
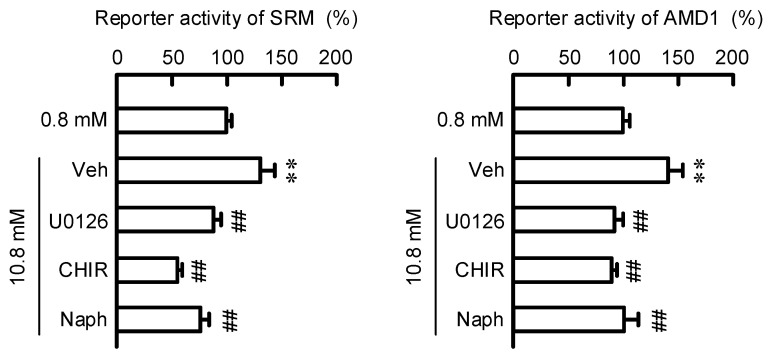
Increases in reporter activities of SRM and AMD1 by MgCl_2_ supplementation. Cells were cotransfected with GLuc-ON Promoter Reporter vectors containing the promoter region of *SRM* or *AMD1*, and SEAP vector. After 48 h of transfection, the cells were pre-incubated in the absence (Veh) and presence of each inhibitor for 0.5 h, followed by incubation with 0.8 and 5.8 mM MgCl_2_ for 6 h. The reporter activities are represented as a percentage of 0.8 mM MgCl_2_. n = 4–5. Error bars indicate SEM. ** *p* < 0.01 significantly different from 0.8 mM MgCl_2_. ^##^ *p* < 0.01 significantly different from Veh.

**Figure 7 cells-11-02268-f007:**
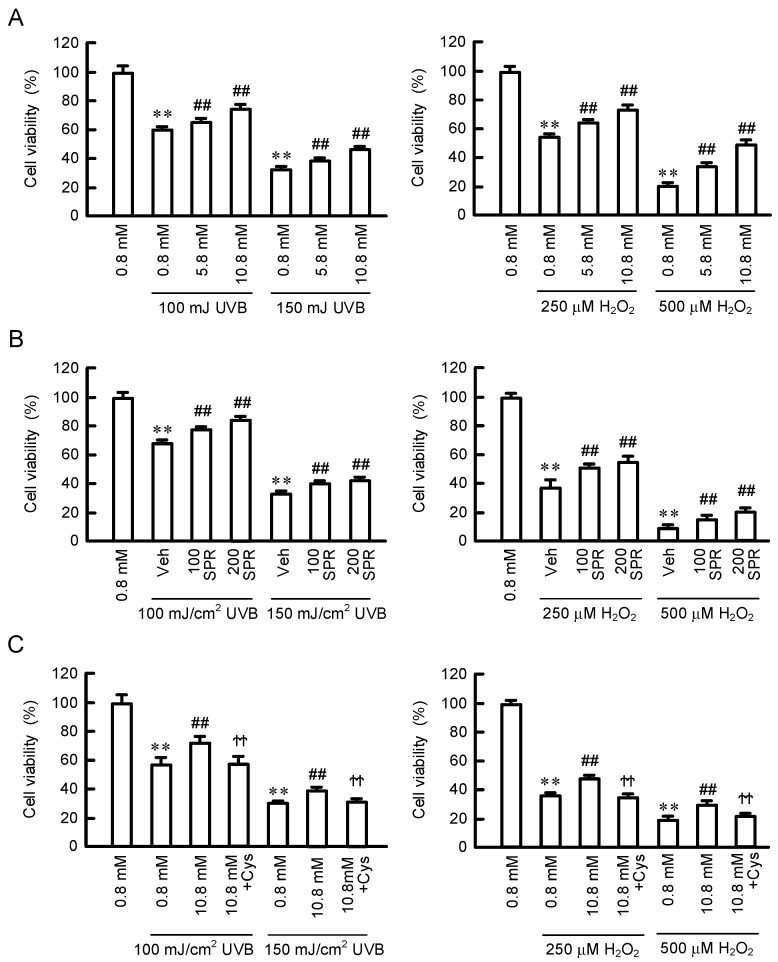
Protective effect of MgCl_2_ supplementation on UVB- and H_2_O_2_-induced cell damages. (**A**) Cells were pre-incubated in the presence of 0.8, 5.8, and 10.8 mM MgCl_2_ for 24 h. (**B**) Cells were pre-incubated in the absence (Veh) and presence of 100 or 200 μM spermidine (SPR) for 24 h. (**C**) Cells were pre-incubated in the presence of 0.8 or 5.8 mM MgCl_2_ and 10 μM cysteamine (Cys) for 24 h. Then, the cells were treated with 100 and 150 mJ/cm^2^ UVB or 250 and 500 μM H_2_O_2_. Cell viability was measured using CCK-8 and represented as a percentage of 0.8 mM MgCl_2_. n = 4–6. Error bars indicate SEM. ** *p* < 0.01 significantly different from 0.8 mM MgCl_2_. ^##^ *p* < 0.01 significantly different from Veh. ^ϮϮ^ *p* < 0.01 significantly different from without Cys.

**Figure 8 cells-11-02268-f008:**
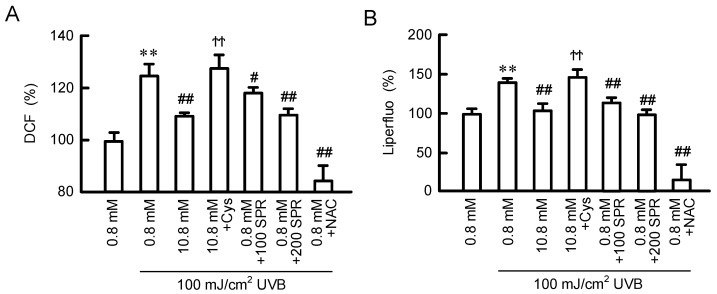
Inhibitory effect of MgCl_2_ supplementation on UVB-induced production of ROS and lipid peroxide. (**A**) Cells were pre-incubated in the presence of 0.8 and 10.8 mM MgCl_2_, 10 μM cysteamine (Cys), 100 and 200 μM spermidine (SPR), or 2 mM NAC for 24 h. Then, the cells were treated with 100 mJ/cm^2^ UVB followed by incubation with 2’,7’-dichlorodihydrofluorescein diacetate (**A**) or Liperfluo (**B**). The fluorescence intensities of DCF and Liperfluo are represented as a percentage of 0.8 mM MgCl_2_. n = 4–6. Error bars indicate SEM. ** *p* < 0.01 significantly different from without UVB. ^##^ *p* < 0.01 and ^#^ *p* < 0.05 significantly different from 0.8 mM MgCl_2_. ^ϮϮ^ *p* < 0.01 significantly different from without Cys.

**Table 1 cells-11-02268-t001:** Primer pairs for real-time PCR.

Name	Direction	Sequence (5′–3′)
*SRM*	Forward	TAGCTCGAAGCTGACCCTACAT
	Reverse	AGAGGACACCATCTTCCTTGAG
*AMD1*	Forward	CAGAGAGTCGGGTAATCAGTCA
	Reverse	CTCTCACGAGTGACATCCTTTG
*β-Actin*	Forward	CCTGAGGCACTCTTCCAGCCTT
	Reverse	TGCGGATGTCCACGTCACACTTC

**Table 2 cells-11-02268-t002:** Lists of genes with change over twofold in DNA microarray analysis.

No.	Gene Name	Gene Symbol
1	*adenosylmethionine decarboxylase 1*	*AMD1*
2	*ATPase family, AAA domain-containing 5*	*ATAD5*
3	*breast cancer 2, early onset*	*BRCA2*
4	*carbamoyl-phosphate synthetase 2, aspartate transcarbamylase, and dihydroorotase*	*CAD*
5	*cell division cycle 6*	*CDC6*
6	*cell-division-cycle-associated 7*	*CDCA7*
7	*centromere protein V*	*CENPV*
8	*chromatin licensing and DNA replication factor 1*	*CDT1*
9	*claspin*	*CLSPN*
10	*clustered mitochondria (cluA/CLU1) homolog*	*CLUH*
11	*coiled-coil domain-containing 86*	*CCDC86*
12	*CTP synthase 1*	*CTPS1*
13	*cysteine-and-histidine-rich-domain-containing 1 pseudogene*	*LOC727896*
14	*denticleless E3 ubiquitin protein ligase homolog (Drosophila)*	*DTL*
15	*DNA replication and sister chromatid cohesion 1*	*DSCC1*
16	*ets variant 4*	*ETV4*
17	*family with sequence similarity 111, member B*	*FAM111B*
18	*general transcription factor IIH subunit 2*	*GTF2H2*
19	*GINS complex subunit 2 (Psf2 homolog)*	*GINS2*
20	*heat shock protein 90kDa alpha (cytosolic), class B member 3, pseudogene*	*HSP90AB3P*
21	*matrix metallopeptidase 1*	*MMP1*
22	*matrix metallopeptidase 7*	*MMP7*
23	*microRNA 1244-1*	*MIR1244-1*
24	*minichromosome maintenance complex component 6*	*MCM6*
25	*minichromosome maintenance 10 replication initiation factor*	*MCM10*
26	*neuropilin (NRP) and tolloid (TLL)-like 2*	*NETO2*
27	*origin recognition complex subunit 1*	*ORC1*
28	*parathyroid-hormone-like hormone*	*PTHLH*
29	*polymerase (DNA directed), alpha 2, accessory subunit*	*POLA2*
30	*polymerase (RNA) III (DNA-directed) polypeptide G (32kD)*	*POLR3G*
31	*pseudouridylate synthase 7 (putative)*	*PUS7*
32	*pumilio RNA-binding family member 3*	*PUM3*
33	*serpin peptidase inhibitor, clade B (ovalbumin), member 3*	*SERPINB3*
34	*serpin peptidase inhibitor, clade B (ovalbumin), member 4*	*SERPINB4*
35	*small ILF3/NF90-associated RNA D*	*SNAR-D*
36	*small ILF3/NF90-associated RNA E*	*SNAR-E*
37	*small ILF3/NF90-associated RNA H*	*SNAR-H*
38	*small nucleolar RNA, H/ACA box 50C*	*SNORA50C*
39	*solute carrier family 43, member 3*	*SLC43A3*
40	*spermidine synthase*	*SRM*
41	*TIMELESS interacting protein*	*TIPIN*
42	*translocase of outer mitochondrial membrane 40 homolog (yeast)*	*TOMM40*
43	*uncharacterized LOC101927746*	*LOC101927746*
44	*VPS9D1 antisense RNA 1*	*VPS9D1-AS1*
45	*zinc finger protein 92*	*ZNF92*

## Data Availability

Not applicable.

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
