# Peer review of "Magnesium Supplementation Attenuates Ultraviolet-B-Induced Damage Mediated through Elevation of Polyamine Production in Human HaCaT Keratinocytes"

_cells, 2022, doi:10.3390/cells11152268_

Round 1

Reviewer 1 Report

In this manuscript the authors shed light to the mechanism of cell protection from UVB of Magnesium. This can be of interest for the scientific community.

The authors conducted several well designed experiments showing that MgCl2 induces the synthesis of polyamines and signaling cascades. Also they proved its protection effect against UVB and  H2O2 insults.

major concerns:

The manuscript has the major limitation that only one cell line was used in the whole study.

The authors performed a DNA microarray comparing normal vs. high concentration of MgCl2. According to their report only 43 genes were induced by high MgCl2. The authors didn't mention those genes downregulated. 

The authors should provide the full list of genes together with their expression values as supplemental material.

A functional analysis, at least with the dysregulated genes, needs to be presented in the results. This will provide a better understanding of the molecular mechanisms of exposure to high concentration of MgCl2. In addition, it might set the basis to explore the effect of Magnesium beyond the scope of this manuscript.

Author Response

We thank you very much for your careful reading of our manuscript and valuable comments.

Comment 1

The manuscript has the major limitation that only one cell line was used in the whole study.

Answer

  Following your indication, we performed additional experiments. The mRNA levels of SRM and AMD1 are also increased by MgCl2 supplementation in NHEK/SVTERT3-5 cells. Please see new figure 2.

Comment 2

The authors performed a DNA microarray comparing normal vs. high concentration of MgCl2. According to their report only 43 genes were induced by high MgCl2. The authors didn't mention those genes downregulated.

Answer

  Following your suggestion, we showed the data of downregulated genes. Please see line 169 and table S2. We checked the upregulated genes again and noticed a mistake. The gene of small ILF3/NF90-associated RNA is divided into three types including D, E, and H. Therefore, we corrected the lists of genes. Please see new table 2 and table S1.

Comment 3

The authors should provide the full list of genes together with their expression values as supplemental material.

Answer

  Following your suggestion, we showed the full list of genes together with their expression values. Please see table S1 and S2.

Comment 4

A functional analysis, at least with the dysregulated genes, needs to be presented in the results. This will provide a better understanding of the molecular mechanisms of exposure to high concentration of MgCl2. In addition, it might set the basis to explore the effect of Magnesium beyond the scope of this manuscript.

My comment was related to a functional annotation enrichment analysis. Basically, what I would like to see in the manuscript is that the authors use the transcriptomic data and submit it to a bioinformatic tool to look for specific pathways altered between the groups. There is a variety of available tools to perform this kind of analysis, such as GSEA, STRING, DAVID bioinformatics, REACTOME, ToppGene, etc. If the authors in addition have siRNA data, they should present it.

Answer

  Thank you very much for your valuable opinion. Following your suggestion, we performed analyses of GO and KEGG. Please see new table S3 and S4. In addition, we discussed the role of high concentration of MgCl2. Please see line 332.

Reviewer 2 Report

In the manuscript by Shu et al., the effect of magnesium supplementation on UVB-induced damage and the underlying signaling mechanisms are explored.  Using the immortalized keratinocyte cell line, HaCaT, they determine that magnesium supplementation increases the expression of polyamine synthases, specifically SRM and AMD1, at the gene and protein levels in a dose-dependent manner. Using various pharmacological inhibitors, the authors conclude that the MgCl2 supplementation appears to elevate SRM and AMD1 via transcriptional regulation by GSK3, CREB, and the MEK/ERK pathway. Importantly, the authors show that UVB-induced reduction in cell viability is rescued by MgCl2 supplementation and that this is dependent on polyamine synthase activity.

Overall, this manuscript reports a compelling role for magnesium supplementation on keratinocyte biology in response to damage induced by UVB. While some of the changes appear somewhat modest, most of the conclusions are supported by the data presented and the interpretations are sound. Including a second keratinocyte cell line, or even primary keratinocytes, would greatly strengthen their findings. 

Most of my major critiques are related to both textual and visual presentation. I feel this manuscript could be greatly improved by providing more thorough descriptions of experimental designs/approaches throughout the Results and Materials and Methods sections. This would assist the readers in following along with the authors’ rationale and help them come to the same conclusions as the authors. More specific critiques are indicated below. 

1.     It might be useful to have a flow-chart or other visual representation of the information provided in the introduction in Lines 55-60 that describes the synthesis of polyamines. 

2.     Can the authors describe why they use the term “keratinocyte-derived HaCaT cells” in the title and throughout the manuscript? Aren’t HaCaT cells human keratinocytes? Maybe they mean “skin-derived keratinocytes”?

3.     The Materials and Methods are unusually sparse. For example, was the DMEM supplemented in any way for culture of keratinocytes? How many cells were plated? In what size dish? How much protein was used in western blots? How much DNA was transfected into cells? Also, catalog numbers should be provided for antibodies and all other key reagents. In general, much more detail can and should be provided. 

4.     In Table 2, please also include gene abbreviations/symbols. In Line 152, the authors refer to gene symbols (AMD1 and SRM), but it is difficult to reconcile with the table that only lists full gene names. 

5.     Throughout the Results sections, the experiments are not described in a manner adequate to assist the reader in understanding the experimental approach/design. For example, in Lines 152-155, the authors simply state that “Real-time PCR analysis revealed that the mRNA levels of SRM and AMD1 are increased by MgCl2 supplementation in a dose-response manner (Fig. 1A)” etc. Was this in HaCaT cells? Are these results from the experiment listed in Lines 148-149, or is that describing the microarray study. This critique applies throughout the Results section. 

6.     Throughout the Results section, the authors do not include any reference to actual data (average values, +/- SEM, statistical comparisons). This would be useful to the reader and would augment their figures, which show somewhat modest effects in some assays. 

7.     In Figure 1, it would be useful to indicate that the 0.8 mM MgCl2-treated cells are “untreated” control cells cultured with the normal [MgCl2] found in DMEM. 

8.     Please indicate what the error bars represent in the Figure Legends. 

9.     In Figure 2B, the IF images are quite blurry. This makes the results less convincing as the PolyamineRED signal appears to have significant background signal that doesn’t appear to be cell-specific. The addition of higher magnification images in this panel would be helpful. 

10.  Discussion: Please refer to individual Figures when they are being discussed. 

Minor: 

1.     Numerous grammatical and sentence structure issues throughout. 

Author Response

We thank you very much for your careful reading of our manuscript and valuable comments.

Major

Comment 1

It might be useful to have a flow-chart or other visual representation of the information provided in the introduction in Lines 55-60 that describes the synthesis of polyamines.

Answer

  Following your suggestion, we described the flow-chart of synthesis of polyamines. Please see new figure 1.

Comment 2

Can the authors describe why they use the term “keratinocyte-derived HaCaT cells” in the title and throughout the manuscript? Aren’t HaCaT cells human keratinocytes? Maybe they mean “skin-derived keratinocytes”?

Answer

  Following your suggestion, we corrected the description.

Comment 3

  The Materials and Methods are unusually sparse. For example, was the DMEM supplemented in any way for culture of keratinocytes? How many cells were plated? In what size dish? How much protein was used in western blots? How much DNA was transfected into cells? Also, catalog numbers should be provided for antibodies and all other key reagents. In general, much more detail can and should be provided.

Answer

  Following your suggestion, we rewrote the Materials and Methods section in detail.

Comment 4

In Table 2, please also include gene abbreviations/symbols. In Line 152, the authors refer to gene symbols (AMD1 and SRM), but it is difficult to reconcile with the table that only lists full gene names.

Answer

  Following your suggestion, gene symbols were added in new table 2.

Comment 5

Throughout the Results sections, the experiments are not described in a manner adequate to assist the reader in understanding the experimental approach/design. For example, in Lines 152-155, the authors simply state that “Real-time PCR analysis revealed that the mRNA levels of SRM and AMD1 are increased by MgCl2 supplementation in a dose-response manner (Fig. 1A)” etc. Was this in HaCaT cells? Are these results from the experiment listed in Lines 148-149, or is that describing the microarray study. This critique applies throughout the Results section.

Answer

  Following your suggestion, we rewrote the Results section.

Comment 6

Throughout the Results section, the authors do not include any reference to actual data (average values, +/- SEM, statistical comparisons). This would be useful to the reader and would augment their figures, which show somewhat modest effects in some assays.

Answer

  Following your suggestion, we rewrote the Results section.

Comment 7

In Figure 1, it would be useful to indicate that the 0.8 mM MgCl2-treated cells are “untreated” control cells cultured with the normal [MgCl2] found in DMEM.

Answer

  Following your suggestion, we modified the figure. Please see new figure 2.

Comment 8

Please indicate what the error bars represent in the Figure Legends.

Answer

Following your suggestion, we added the explanation of error bars in the figure legends.

Comment 9

In Figure 2B, the IF images are quite blurry. This makes the results less convincing as the PolyamineRED signal appears to have significant background signal that doesn’t appear to be cell-specific. The addition of higher magnification images in this panel would be helpful.

Answer

Following your suggestion, we replace the representative images. Please see new figure 3B.

Comment 10

Discussion: Please refer to individual Figures when they are being discussed.

Answer

Following your suggestion, we referred to individual figures in the Discussion section.

Minor

Comment 1

Numerous grammatical and sentence structure issues throughout.

Answer

Thank you for pointing out. We have carefully checked again.

Round 2

Reviewer 2 Report

The authors have adequately and appropriately responded to my original critiques. They have amended and/or added updated text and figures that I feel improve the clarity of their presentation. There are still some grammatical issues throughout that should be corrected prior to publication, but these do not detract from the overall improvements made during the revision process. 

Author Response

We thank you very much for your careful reading of our manuscript and valuable comments.  Following your indication, we checked the grammatical mistakes in the manuscript and made corrections.